# GoTO: A Process-Navigation Tool for Telehealth and -Care Solutions, Designed to Ensure an Efficient Trajectory from Goal Setting to Outcome Evaluation

Lars Kayser *[ID], Dorthe Furstrand [ID], Emil Nyman Rasmussen, Ann-Catrine Monberg and Astrid Karnoe

Section for Health Services Research, Department of Public Health, University of Copenhagen, DK-1353 Copenhagen, Denmark
* Correspondence: lk@sund.ku.dk

**Abstract:** Objectives: The digital transformation of the healthcare sector involves the procurement and implementation of new health technologies, which will likely be a challenge to healthcare providers who are not part of large organizations. In response to the needs of small and middle-sized health and care organizations, we have developed a process navigator to guide providers of healthcare through the processes of innovation, the procurement of mature products, and their implementation in telehealth and telecare projects. Methods: A narrative overview identified health-technology-assessment-inspired models. Conversations with national and international colleagues identified project and implementation models. The origin of the included models was identified, and relevant articles were referred to to describe the essential principles, including the nature of stakeholder involvement and the evaluation processes when appropriate. Based on the inputs, we proposed the process navigator GoTO. Results: Six health-technology-assessment-inspired models, six project models, one implementation model, and one innovation model were identified and informed the creation of the GoTO process navigator. The navigator consists of four parts: inception (eight steps); materialization (three tracks, depending on the maturity of the planned solution); implementation (five steps); and the final assessment and evaluation. Conclusion: The GoTO process navigator is an intuitive guide for innovation, procurement, and implementation in telehealth and -care. The GoTo navigator can assist providers of digital health and care services throughout the process from the initial identification of goals to the final evaluation of outcomes.

**Keywords:** health technology assessment; project model; implementation; telehealth; telecare

## 1. Introduction

Around the world, it is recognized that people with health issues need to be more involved in their own health and become more health-literate and empowered, as this will ultimately increase their quality of life and well-being [1–3]. More than a decade ago, it became evident that home telemonitoring or web-based interventions were able to empower people by increasing their insight into their own health conditions, their self-efficacy, and, ultimately, their ability to manage their own health conditions [4,5].

During the last decade, there has been a rapid dissemination of smartphones, and today there are more than 3.6 billion users on social media platforms such as Twitter, Facebook, and Weibo worldwide. This widespread use and adoption of smartphone technologies provides an opportunity to connect patients to healthcare providers and link them in a collaborative effort to increase empowerment and self-management [4–8]. Here, it should also be noted that the rapid development of everyday technologies (e.g., smart homes and wearables) has increased expectations among consumers and providers of health and care services for access to more agile and technology-enabled interactions and support in relation to health care [5,9].

In response to the increasing demand for technology-enabled interactions, providers of health and care services are increasingly engaged in the innovation, procurement, and implementation of telehealth and -care (THC) solutions. This can be a complex area to enter [10] and represents a significant challenge for many healthcare providers. In the process from innovation to implementation there are potential pitfalls. Large organizations may be able to tackle these challenges by relying on frameworks and models such as the intervention map framework [11], the Medical Research Council (MRC) guidance for complex intervention [12,13], the Epital care model [6,14], or by setting up a project-management organization based on the certification of the people involved, such as PRINCE2 [15]. Small and middle-sized health and care organizations (SMHOs), however, often have less resources, which may result in a lack of awareness of previous results and an unwillingness to face and tackle organizational challenges; therefore, these organizations may find it difficult to initiate and implement new solutions [16,17].

Over the years, many SMHOs, such as hospitals, municipalities, and counties, have needed to identify tools and frameworks to assist the process from the innovation to implementation and advice on how to implement THC solutions using these frameworks successfully. In the United Kingdom, the West Midlands toolkit (WMT) was developed to address this challenge [18].

A project mapping the Danish telemedical landscape provides an example of how THC projects still struggle with navigating the process from innovation to implementation. The mapping project identified more than 350 local Danish THC projects [19], but, as we have documented previously, only a few of these projects were ever evaluated or had their results communicated to the public [16]. Today, abundant literature exists documenting the effects of telehealth solutions and how they can be implemented [20,21]. In spite of these rich resources with many different approaches, SMHOs still require access to a process navigator that can help them overcome the challenges they face and avoid running into known barriers such as the inadequate consideration of clinical context, workflows, and communication [21] or the lack of organizational coherence, cognitive participation, and collaborative action [22].

In particular, the past year's challenges during the SARS-CoV-2 pandemic highlight the need for and benefits of innovative THC solutions and underline the necessity of efficient and effective ways to implement these solutions. New resources and insights that can assist and guide in making innovation and procurement in the increasing digitalized health sector simpler and more intuitive for health and care providers are therefore needed.

To meet these challenges, our objective was to create a process navigator that can serve as a tool to guide and assist providers of health and care services in their innovation and procurement of THC solutions, covering the full range from the initial identification of goals to the final evaluation of outcomes. The process navigator was named GoTO (Goal to Outcome) because it helps the actors involved navigate through the identification of goals, ensures awareness and ownership among the actors involved, and supports the alignment of steps and processes with a trajectory towards the expected outcomes.

Based on our experiences prior to this study, we believed that certain steps and processes necessary to achieve successful innovation and procurement could be identified as common across the procurement and implementation models used in the provision of health and care, and that this information could be condensed and used to develop a simple model to guide these processes.

Our initial assumption was that health technology assessment (HTA)-inspired models possess essential key characteristics such as evidence, structure, and a socio-technical approach, and that essential steps could be identified by aligning and comparing them to each other. HTA is defined by the WHO as "the systematic evaluation of properties, effects, and/or impacts of health technology" [23] and uses a socio-technical approach addressing the social, economic, organizational, and ethical issues of health technology as well as the impact of health technology on users [23]. It is essential to include and address

these aspects in projects, similarly to how HTA has already inspired new models for the acquisition of hospital or telemedicine products [24–26].

In the synthesis of the literature, we realized that to develop a complete model that could navigate SMHOs through all processes in the development and implementation of THC solutions, we needed to widen the scope and include key components from project and innovation models. To accommodate this need, we identified commonly used healthcare project models suitable for this purpose in dialogue with national and international experts.

The aim was to explore what the key components in existing HTA-inspired models are and how these key components can be combined to inform a process navigator.

To address this aim, we first conducted a narrative literature overview and then combined the result with an analysis of widely accepted innovation and project models in healthcare.

## 2. Methods

The study consisted of three stages. Stage 1 was a narrative overview based on a literature search for HTA and related models used with regard to THC projects. The narrative overview was part of author ACM's master's thesis, which is unpublished, and resulted in a preliminary model based on HTA. Based on the thesis, we continued the development process of the GoTO model in stage 2 by reviewing and revising the material from the thesis and expanding the scope to include project management, development, and implementation models that had proven successful in relation to healthcare. In stage 3, we discussed the findings internally and with colleagues in Norway, Denmark, and Australia. Based on a summary of these discussions and our experience, we proposed the GoTO process navigator.

In stage 1, the narrative overview of HTA models was conducted as recommended by Green et al., 2006 [27]. A narrative overview is a type of literature review that, in contrast to a systematic qualitative or quantitative review [28,29], has a less systematic approach, as it involves primarily identifying literature in relevant databases and then complementing these findings with a more unsystematic supplement of grey literature or suggestions obtained through dialogues with experts. Therefore, the result is a synthesis based on a combination of literature and expert insights, which can be used to elaborate on [27].

Initially, a literature search was performed using PubMed to identify peer-reviewed articles in the area of THC HTA-related models in March–April 2014 (ACM, LK) and was limited to the preceding decade (2003–2013). The search terms used were created based on the aim of the project: "(Quality OR telecare OR telemedicine) AND (implementation) OR tool OR innovation) AND (evaluation OR technology assessment) AND HTA" and "Telemedicine AND HTA". We planned to exclude any articles with full text in languages other than English or Danish, but none were identified. On this basis, we found 130 articles.

The identified articles were screened by title for relevance of inclusion, and, in some cases, the abstracts were read to ensure the inclusion of only relevant articles prior to reading the full text for final inclusion. Articles including HTA and telecare, telehealth, or telemedicine were used if they either described a specific model inspired by HTA or they reported on a model's usage in relation to procurement or implementation. In total, six articles were found, three of which described the same model, resulting in the identification of four models: the EUnetHTA framework, mini-HTA for hospitals (mini-HTA), the model for assessment of telemedicine (MAST), and the constructive technology assessment (CTA) (see Figure 1) [24,26,30,31].

The identified literature was supplemented with input from subject-matter experts contacted directly, which resulted in the identification of the continuous and systematic evaluation (CSE) model by Catwell et al. [32]. In addition, experts in Norway introduced us to Masella and Zanaboni's research and resulting model for acquisition, which we here refer to as the decision-making model (DMM) [33].

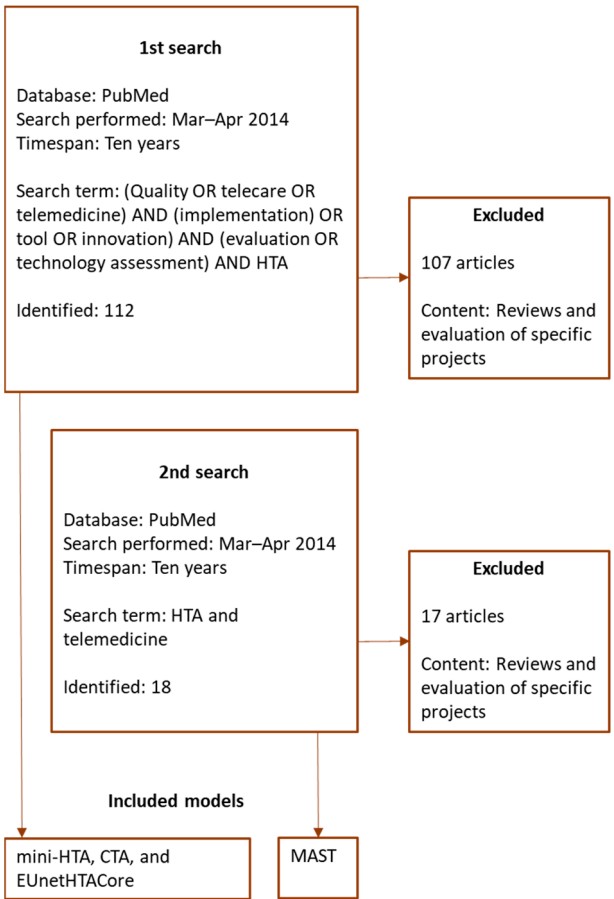

**Figure 1.** Overview of literature search.

For the purpose of the master's thesis, ACM and LK selected two other models with a focus on development and implementation in a health context in addition to the previously identified HTA-inspired models in order to propose a preliminary model: Stanford's Biodesign and the West Midlands toolkit [18,34].

In stage 2, our research group continued the development process in 2015–2016. Authors AK and LK revisited the above-mentioned models and prepared a synthesis of the proposed framework for publication that was shared with the whole author group. However, in this process, we identified that the synthesis of key components lacked elements that could guide the management of processes from innovation to implementation. Therefore, we decided to supplement the above-mentioned six models with project models to understand the essential principles of a successful user-involved, agile-assessment-based process. The project management models were identified together with national and international colleagues working in the field. This resulted in the identification of six models.

In stage 3, we created the GoTO process navigator. This was based on the "benefit trajectory" model [35] that was initially proposed in the thesis and originally consisted of three parts. The revision of the model built on a critical examination of the articles in stage 1 and a synthesis of these findings and the addition of the insight from stage 2. This resulted in a four-part model with additional elements. These elements were based on our findings in stage 2 and theories of normalization processing and person-centeredness. The model needed to be person-centered, as this was the key concept of most THC solutions and the strategies of WHO [1].

To align with the normalization process theory and the WMT-based experiences, the model needed to comprise several steps to ensure the ownership of all stakeholders [18,22]. Based on what we had learnt from CSE, PRINCE-2, and the MAST model, we found

it important to include both iterative processes with formative evaluations and a final assessment of the outcome. We also judged that these steps should be included in one or more of the parts. In the following section, we present the results of stage 2 and stage 3.

## 3. Results

We here present our findings for: (1) six HTA-inspired models, (2) six project management models, and (3) two development and implementation models.

### 3.1. HTA-Inspired Models

In this section, we will describe the key characteristics, similarities, and differences of HTA-inspired models, see Table 1. Two of the models, CTA [30] and CSE [32], are primarily formative and focus on quality, safety, and efficiency as well as the process of moving from development in confined areas to more general deployments. The four other models, EUnetHTA [31], mini-HTA [36], MAST [24], and DMM [37], are intended for decision making regarding the acquisition of new technologies. These models focus primarily on summative evaluations of mature technologies. Maturity is considered a requirement to perform a valid HTA [24]. This is evaluated by criteria of sufficient clinical evidence and/or if the technology has passed the prototype and proof-of-concept phases. DMM differs from the other three models by being a two-step model primarily designed for procurement.

### 3.2. Project Management Models

In this section, we present six project management models including simple linear models, such as the waterfall model; iterative models, such as plan do check act, the agile manifesto, and scrum; and complex models, such as PRINCE2 [40–43]. See Table 2 for an overview of the identified project management models.

Understanding the principles of these project management models contributed information that enriched the formative HTA-inspired models, such as CTA for health IT and CSE [30,32], presented in the previous section.

### 3.3. Development and Implementation Models

In this section, we present an implementation model that has been used for THC in the United Kingdom for several years. This model adds to the principles of management and addresses communication and dissemination. We also include an innovation model from Stanford University that centers on translating problems to needs and focuses on economic sustainability and growth from an industrial perspective.

#### 3.3.1. West Midlands Toolkit

The WMT was developed as part of the West Midlands telehealth project in 2009 [48] and was updated to version 2.0 in 2012 [18]. The WMT is an eight-step innovation and deployment model developed specifically for telehealth and telemedicine to provide a framework for decision makers. The steps are: 1. identifying needs; 2. establishing buy-in; 3. considering technology; 4. workforce requirements; 5. designing evaluation; 6. planning implementation; 7. business case; and 8. sharing best practice. These eight steps can be divided into three phases. Steps 1–4 constitute the inception phase, where needs, current practice, and existing technology are evaluated in order to create a new solution. Steps 5–6 cover implementation design, including milestones with formative evaluation points; step 7 focuses on the verification of the business case; and step 8 leads to an extensive plan for knowledge dissemination.

To support transferability to other project settings, the WMT was provided as an interactive toolkit that can be downloaded [18].

**Table 1.** An overview of the six identifed HTA-inspired models.

| Name and Origin | Objective | Components |
|---|---|---|
| Constructive technology assessment for health information technology (CTA). CTA for health IT was developed in the late 00s based on the CTA described by the Netherlands Organization of Technology Assessment (NOTA) in 1987 [30,38]. | To handle the complexity and varying needs of the healthcare sector, offering agile implementation and formative evaluation to enable an adaptive implementation of new technologies. | Five stages: 1) Research and planning; 2) Design; 3) Development; 4) Implementation and diffusion; 5) Summative evaluation and reporting. |
| A continuous systematic evaluation model (CSE), proposed by Catwell et al. in 2009 [32] and based on a seven-step implementation model proposed by Thorley [39]. | A continuous, systematic evaluation of e-health projects to ensure quality, safety, and efficiency. | Four phases: 1) Inception; 2) Requirements and analyses; 3) Design, develop, and test; 4) Implement and deploy. |
| Mini health technology assessment for hospitals (Mini-HTA), proposed by the Danish National Board of Health in 2005 [26,36]. | A time-effective model to be used in hospitals by managers in decision making when new technologies are being considered. It is evidence-based and cross-disciplinary. | Four dimensions: 1) Technology; 2) Patient/citizen; 3) Organization; 4) Economy. |
| EUnetHTA framework (EUnetHTA), developed in 2006–2008 by the European network for HTA [31]. | To provide a glossary and tools, including lists and additional resources, to ensure the relevance, reliability, and transferability of data and information from existing HTA reports and to identify areas in need of further development. | Nine dimensions for assessment: 1) Health problem and current use; 2) Description and technical characteristics; 3) Safety; 4) Clinical effectiveness; 5) Costs and economic evaluation; 6) Ethics; 7) Organizational aspects; 8) Social aspects; 9) Legal aspects. |
| Model for assessment of telemedicine (MAST), developed by Kidholm et al. in 2009 and commissioned by the European Commission [24]. | A framework to assist organizations in deciding whether a specific telemedical technology is suitable for implementation. | Three steps: 1) Preceding considerations; 2) Multidisciplinary assessment (seven dimensions); 3) Assessment of transferability. The seven dimensions in step 2 are: health and technology aspects, safety, clinical effect, patient perspective, economics, organization and sociocultural aspects, and ethics and law. |
| Decision-making model (DMM), proposed by Zanaboni et al. in 2011 [37]. | To support decision makers in the acquisition of scalable telemedical solutions. The intention of the model was to evaluate solutions or technologies presented by vendors in a competing process. | Two stages: 1) Assessment of preliminary proposals; 2) Assessment of full proposals by multidisciplinary boards of experts. |

### 3.3.2. The Stanford BioX Biodesign Innovation Process

Before discussing the HTA-related and project models identified by our search, it is important to introduce the Stanford BioX Biodesign innovation process [34], as it is complementary to the other models. Biodesign is suitable when the project starts by focusing on problems and gaps, where other projects, given the nature of HTA, start with a focus on technology. Biodesign comprises a pre-project phase involving the establishment of a team and the strategic focus of the work, an identification phase, an invention phase, and an implementation phase. During these four phases, an iterative process takes place to ensure the relevance of the project in relation to existing products and clinical needs and to understand its potential impacts on stakeholders.

**Table 2.** An overview of the six identified project management models.

| Name and Origin | Objective | Components |
|---|---|---|
| Plan do check act (PDCA). Originates from Deming in 1950 [40]. | Intended as a problem-solving model. Ensures development and implementation processes that lead to a product suitable for the market. | Four steps: <br> 1. Plan—plan the project. <br> 2. Do—execute the plan. <br> 3. Check—perform analysis or evaluation. <br> 4. Act—act on evaluation results, adjust the plan, and if needed repeat the cycle. <br><br> In the 1960s, Ishikawa enhanced the model for the purpose of implementation by adding 'goal description' to the plan phase and 'educational activities' to the do phase. |
| The lean approach (LEAN). The term 'lean production' was first introduced in 1990 by Womack, Jonas, and Ross [44]. | A production system that produces more and better products using less time, less space, and fewer labor hours. The goal of LEAN is to deliver the product while maximizing value and minimizing waste in the production process. | Five overlapping phases: <br> 1. Project definition; <br> 2. Lean design; <br> 3. Lean supply; <br> 4. Lean assembly; <br> 5. Use. <br><br> Learning loops are incorporated between all phases to ensure the transfer of new experience. |
| The waterfall model. The term 'waterfall model' was first introduced in 1976 by Bell and Thayer [45], based on Royce's conceptualization of H.D Bennington's model for software development [43]. | An intuitive linear approach with steady requirements, most suitable for mature and stable environments. | Seven steps: <br> 1. System requirements; <br> 2. Software requirements; <br> 3. Analysis; <br> 4. Program design; <br> 5. Coding; <br> 6. Testing; <br> 7. Operations. |
| Scrum. Described by Nonaka and Takeuochi in 1986 [46], introduced for object-oriented development in 1995 and described as an agile methodology in 2001 [41,47]. | An agile method for software development based on key characteristics identified in successful companies. | Three phases: <br> 1. Planning; <br> 2. Sprint; <br> 3. Closure. <br><br> The first and last phases—planning and closure—are well-defined, involving the specification of input and output. The flow is linear, and there may be a number of iterations during the planning phase. The sprint phase is a nonlinear and highly flexible phase. The project is governed by a project owner and involves a backlog. |
| PRojects IN Controlled Environments (PRINCE-2). Named PRINCE-2 in 1986, derived from PROMT used by the Central Computer and Telecommunications Agency since 1979 as the standard to be used for IT projects [15]. | The method is based on seven principles, with a project involving four stages, seven processes, and seven themes, which make it possible to tailor the PRINCE2 method to any size or type of project. | Four stages: <br> 1. Pre-project; <br> 2. Initiation stage; <br> 3. Subsequent delivery stage(s); <br> 4. Final delivery stage. <br><br> The seven principles are: continued business justification, learn from experience, defined roles and responsibilities, manage by stages, manage by exception, focus on products, and tailor to suit the project environment. <br> The seven processes are: starting up a project, initiating a project, directing a project, controlling a stage, managing product delivery, managing a stage boundary, and closing a project. <br> The seven themes are: business case, organization, quality, plans, risk, change, and progress. |

### 3.4. Summary

In this section, we present a brief summary of the key components and elements that informed the creation of the GoTO process navigator.

The identified HTA-related models ranged from very rigid models (mini-HTA) to more agile and flexible approaches (CTA), which was also reflected in the chosen principles

of evaluation, with the summative approach in mini-HTA and MAST and formative evaluation in CTA for health IT and CSE. Additionally, the way the users and organizations are involved in the process varies from almost co-creation dialogues, such as in CTA [30] and CSE [32] which both directly involving the end-users, to a data reporting form, such as EUnetHTA [31].

The project and implementation models also reflect a change overtime moving from software development in linear structures (waterfall), to today where the newer models reflect the increased complexity of projects (agile, PRINCE2). The PDCA model is positioned between these two approaches by being a relatively linear process that can run iteratively in settings that are more complex.

The WMT is distinguished from the other models by having been developed specifically as a project or implementation tool aimed at supporting changes in the healthcare sector. The model was developed as a toolkit to provide support for local organizations, and it is mainly based on a waterfall-like format, but with a PDCA structure in its implementation stage.

## 4. Creation of the GoTO Process Navigator

The GoTO process navigator is presented in Figure 2 and consist of four parts: Part 1—Inception, Part 2—Materialization, Part 3—Implementation, and Part 4—Final Assessment and Evaluation.

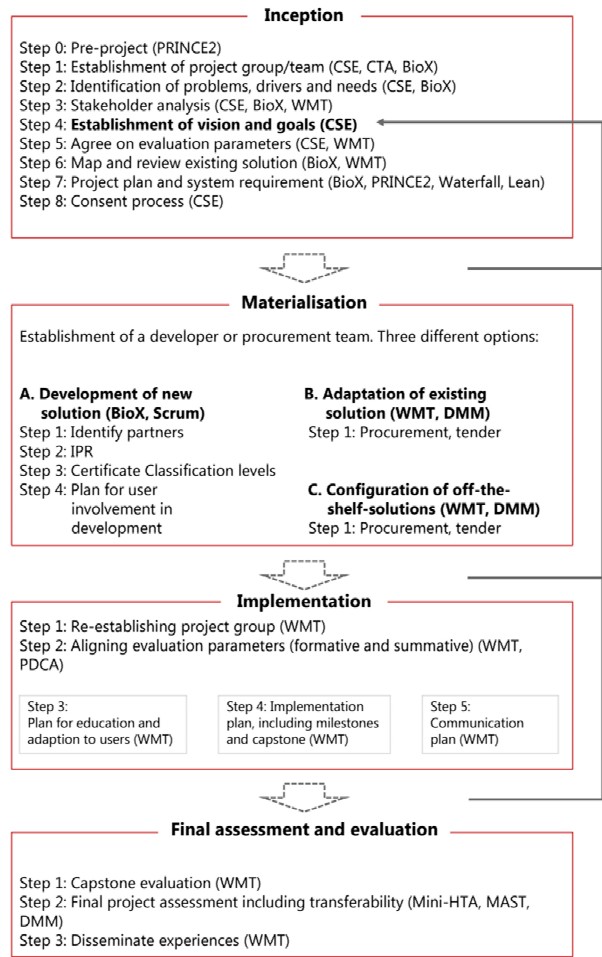

**Figure 2.** Overview of the Goal-to-Outcome (GoTO) model with its four parts. The identified models that informed the design of each of the four parts are indicated. For an overview of the abbreviations used, see Abbreviations.

### 4.1. Part 1—Inception

Part 1 of the GoTO model, "inception", builds on the HTA-inspired models and the Stanford BioX Biodesign innovation process. This part ensures ownership by the leadership, the involvement of relevant stakeholders, and the important definition of a realistic common goal.

The inception part begins with the establishment of a working group, which according to CSE [32] should not refer to, or directly include as members, the leaders at organizational levels. However, it is important that in a pre-project stage, a mandate has been created expressing the leaders' ownership. The eight steps of the inception part ensure that insight is gained into the assumptions behind the project, including clearly defined problems to be solved, the relevance of the project, and the impact on stakeholders; they also ensure that the vision and goals of the planned project are clear, transparent, and based on the state of the art. The inception part concludes with a reality and feasibility check to ensure that the resources and budget will be durable and that the project will be able to proceed to the development or procurement of a solution with established evaluation parameters that are relevant to the goal and the solution described in the requirement specification.

It is important that the alignment between steps five and eight, concerning the types of evaluation parameters used and their establishment, is discussed both during and after the project. This discussion should include human, technological, organizational, and economic dimensions to generate results that can be documented. The most important issue is to agree on a few key parameters for the final assessment that fully align with the vision and goals to serve as targets when following the project trajectory.

### 4.2. Part 2—Materialization

This part builds on three approaches informed by the BioX design model [34], the scrum model, the WMT [18], and the DMM [37], in addition to the literature mentioned below.

In part 2 of the GoTO model, "materialization", one of three approaches can be selected: (A) development, either in the organization or with external partners; (B) procurement, with adaptation to the organization's needs; or (C) the procurement of an off-the-shelf solution.

The materialization step starts with the selection of the approach and ends with the acquisition of the solution by the organization based on a contract or an agreement.

The three paths or approaches a project can follow for materialization differ with respect to complexity. The developer path may simply involve development together with a well-tuned in-house facility, or it may entail research-based or company collaboration. If a project manager finds herself moving in this direction, she needs to consider whether this path really is the right one, or whether the project lacked the adequate mapping and review of existing solutions. If an in-house resource exists, the project manager should team up with them for this part. If the project appears to be research-driven or a company collaboration, the project leader should gather a team for this purpose and consider drawing on external assistance during this part.

When adapting an existing solution, the project manager should involve local resources to identify any need to integrate existing systems or use standardized data formats. This is often a part of the system requirements as well. In regions with mature digital health services, reference architectures documenting how to ensure interoperability may exist [49]. The adaptation often addresses changes in organizational roles, functions, and cultural aspects to increase the likelihood of the adoption of the new solution. Any required add-ons may need to be developed and/or tested in collaboration with end-users [50].

If the path follows an off-the-shelf solution, the product is likely to be documented, and the main concern will be whether the context is new and to what extent educational programs already exist. This information must be included in the implementation part.

For all three approaches, experts in usability, health literacy, or e-health literacy should be included to ensure that these areas are properly addressed and improve the user experience and adoption of the product [51,52]. It is also important to ensure that a THC solution

meets the legal medical classification requirements and that data will be stored according to existing rules, such as the EU's General Data Protection Regulation.

### 4.3. Part 3—Implementation

Part 3 of the GoTO model, "implementation", built mainly on the WMT [18] but was also informed by the need for formative evaluation with iterative adjustments using PDCA cycles [53].

Implementation is a combined effort between the project organization and stakeholders. At this point, a product is ready for implementation and adoption by the users. Through the implementation process, the need to adapt the product or service to ensure its value will naturally arise. The process starts all over again with the identification of the emerging needs. This stage is characterized by being both iterative and formative, and the involvement of users through, for example, user-centered or participatory design [51,54,55] ensures an optimal positive impact via ongoing assessment coordinated by the project leader and adaptation throughout the implementation process. When the initiation phase has been planned and carried out satisfactorily, the next implementation stage starts with planning.

Implementation is often complex and is divided into tasks. These tasks should be clearly described to explain how they are linked together and placed in the trajectory of the project. A PDCA approach is applied to each of these tasks. After deployment, when users' needs have been secured and the technology is being used in an optimal manner to support these needs, it is time to continue to part four, the summative evaluation of whether the expected outcomes have been met.

### 4.4. Part 4—Final Assessment and Evaluation

This part is informed by the consideration of summative evaluations and the description of outcomes using HTA-inspired categories, as described in MAST [24,26].

The final assessment should be true to the vision and goals of the project and be conducted in accordance with the evaluation plan agreed on in part three, including both milestones and the capstone evaluation. The results, effects, and impact can only be truly reported in the context of the planned intervention and its hypothesized gains. If other results—positive or negative—arise, they should be taken into consideration and evaluated in a new setting or iteration.

The final assessment should clearly state each goal, the anticipated outcome, the planned intervention, and how it was evaluated. Here, the occurrence of unanticipated outcomes should also be addressed. If the anticipated result was not achieved, an explanation should be provided, including (most importantly) suggestions for how the result may be reached in the future or why it should not be pursued any longer. This documentation should be published in a manner that allows others to access the information as planned in part three, communication. This assessment should always be conducted, even if the project stops earlier than planned or the goals are not achieved, as it can help the people involved in the process learn from the project [16].

The proposed GoTO process navigator must be implemented according to the guidance provided for each part and step. However, the headings may serve as a checklist for those responsible for the project; inspiration for how to proceed through the steps can be found in the references.

## 5. Discussion

In this article, we presented a navigator that can be used by SMHOs such as municipalities, hospitals, or patient organizations to develop and implement THC solutions in response to identified needs relating to telehealth consultations, such as screen consultations or apps to support communication in relation to a clinical path. In 2020–2022, this was of particular interest due to the increased focus on physical distancing worldwide during the pandemic.

The process navigator was named GoTO (Goal to Outcome) because it helps the actors involved navigate through the identification of goals, ensures awareness and ownership among the actors involved, and supports the alignment of steps and processes with a trajectory towards the expected outcomes (Figure 2).

We initiated the development of this tool despite our knowledge of many existing models, as several projects have reported a lack of a clear vision, organization, or evaluation strategy [16]. We applied a scientific approach by building on HTA-inspired models, which involved a socio-technical focus addressing users, competence, economy, and infrastructure, as well as evidence [23]. We were not able to identify a common denominator across all six identified models, but we were able to identify some key characteristics shared between the models, such as formative vs. summative evaluation and differences in the maturity of the implemented technologies. Here, the focus was on the inclusion of formative evaluation as a process for less mature products that are still being developed and summative evaluation and transferability for mature products, which often come off the shelf and may only need minor modifications. Both of these approaches were incorporated into the GoTO process navigator to allow for both development and procurement depending on the particular needs, making the model agile.

The literature search conducted in 2014 only identified a limited number of models, which may have been due to our focus on HTA-inspired models. As the main objective was to create a feasible navigator easing the work of the local project manager, we avoided inspiration from theory-based models such as the presented theory-based mini-HTA [26]. We also avoided building on the production of evidence in the projects such as in the intervention map framework [11], as this is not feasible for SMHOs. In contrast to our expectations, we were not able to develop the model without including sources other than the literature review of HTA-inspired models.

The main idea of the GoTO was built on CSE [32]. The process navigator provides more details and expands on CSE by changing the focus from continuous evaluation to the processes and how to align all efforts from the initial definition of goals to achieving the expected outcomes.

Despite the existence of the models identified in this paper, projects continue to develop their own innovation and evaluation models [16,19]. This may be due to either a lack of knowledge of the tools, models, and frameworks available, or, more likely, the fact that new principal investigators or persons responsible for the innovation or procurement of new technologies in the area of THC find the existing models too difficult to understand or unsuitable.

By introducing the GoTO process navigator, we hope to provide these organizations with an intuitive and simple model to be used for the procurement and/or innovation of THC solutions, reducing the time many now spend on reinventing the wheel and saving costs by obviating the need to hire external consultants when a new project is initiated.

A strength of the GoTO process navigator is the dynamic inclusion of stakeholder involvement and the identification of their needs in part 1 and part 3. This is in alignment with the ongoing transformation of healthcare services and the increasing involvement of patients or citizens as co-creators in these projects.

The nature of the narrative overview was one limitation of this study. The objective was to identify only HTA-inspired models, as our initial assumption was that these would systematically address the socio-technical aspects of the health context, including economic and ethical aspects. This excluded the opportunity to identify other models that were used outside the health domain or were less assessment-oriented.

The use of "HTA" as a search term in our PubMed search instead of "Health technology assessment" narrowed our search more than intended, and therefore we may have missed frameworks or models other than those identified in the review. Consequently, we may have overlooked concepts or ideas that could have further contributed to elements of our model. This may also explain why we were able to identify additional frameworks through consultations with our colleagues.

The identification of additional project and innovation models as well as the conversations with experts compensated for this limitation, but only to a certain extent. We fully acknowledge the existence of models and frameworks that could have informed our work and recognize that we did not implement clear inclusion and exclusion criteria with respect to what kind of models and frameworks we included in stage 2. One criterion was that the models had been used in relation to telehealthcare projects, but this does not explain why others were not included, e.g., those working with determinants [11–13].

In spite of this, we are confident that the proposed GoTO model is a condensation of the most important areas of project navigation in a THC context. Another limitation may be that the model does not address the steps prior to mandating a project. This area is often related to the business model of an organization. This has previously been covered by Ward and Daniel [56]. On the other hand, the GoTO model is a technology-oriented and more detailed process navigator that will assist SMHOs and their project leaders with a practical navigator. The GoTO tool can also act as an "extension" to Ward and Daniel's model of benefit management and realization [56].

We still need to document the efficiency and effectiveness of the GoTO process navigator when applied to a project, but we benefitted from it in our ongoing work planning studies and educating students at the master's level in health informatics at the University of Copenhagen. The GoTO tool was also used as a navigator for a recently funded EU–Canadian project, the SMart Inclusive Living Environments (SMILE) project, where the aim was to design and implement a conversational agent and a digital care facilitator in four living labs in Canada, Denmark, Norway, and the Netherlands [57].

In 2017, a Danish national network aimed at supporting the development and application of e-health solutions used the GoTO navigator as a template for a toolkit to inform their members. For each of the inception and materialization steps, resources were provided, including links to regulatory rules and interviews documenting facilitators and barriers. Here, the network organizers found the GoTO navigator to be a valuable way to organize and present relevant information [58].

## 6. Conclusions

In this study, we developed the GoTO process navigator, which offers intuitive and simple guidance to the providers of health and care services in their innovation and procurement of THC solutions, from the initial identification of goals to the final evaluation of outcomes.

Although it still needs to be evaluated, it is anticipated that the GoTO process navigator will be an important tool for small and middle-sized organizations providing healthcare and services and their project leaders, offering a way to avoid the initiation of projects that are doomed to fail due to a lack of clear goals and evaluation plans. The availability of the process navigator may also facilitate processes involving radical and market-creating solutions that challenge existing structures, since the inception part encourages processes such as disruptive or catalytic innovations that challenge an institution's organization and business structure.

**Author Contributions:** The idea was initially conceived by A.-C.M., who wrote a master's thesis proposing an initial model. L.K. and D.F. wrote the first draft, A.K. wrote the second draft together with L.K. and E.N.R. participated in describing the project models. The final model was produced in collaboration between L.K., A.K. and D.F. with advice from A.-C.M. All authors contributed to the final revision of the manuscript and have read and agreed to the published version of the manuscript.

**Funding:** L.K. has been partly funded during the revision in 2021 and 2022 by the European Union's Horizon 2020 Research and Innovation program under Grant Agreement n° 101016848 (CIHR), which has also supported the publication.

**Institutional Review Board Statement:** Not applicable.

**Informed Consent Statement:** Not applicable.

**Data Availability Statement:** The 'Methods' section includes the search strings. The results of the searches can be obtained from ACM, as these data were part of her master's thesis.

**Acknowledgments:** Lena Sundby Jensen is thanked for assisting L.K. during the writing of the first draft. Emily Duminski and Astrid Laura Dam Jensen are thanked for their assistance in the preparation of the manuscript for publication.

**Conflicts of Interest:** The authors declare no conflict of interest.

## Abbreviations

| | |
|---|---|
| CSE | Continuous systematic evaluation |
| CTA | Constructive technology assessment |
| DMM | Decision-making model |
| GoTO | Goal to Outcome |
| HTA | Health technology assessment |
| MAST | Model for assessment of telemedicine |
| MRC | Medical Research Council |
| PDCA | Plan do check act |
| PRINCE | Projects in controlled environments |
| SMHO | Small and middle-sized health and care organizations |
| THC | Telehealth and telecare |
| WMT | West Midlands toolkit |

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
