# Peer review of "GoTO: A Process-Navigation Tool for Telehealth and -Care Solutions, Designed to Ensure an Efficient Trajectory from Goal Setting to Outcome Evaluation"

_informatics, doi:10.3390/informatics9030069_

Round 1

Reviewer 1 Report (Previous Reviewer 2)

Thank you for addressing my comments. The revision has improved the manuscript substantially. I agree with the authors that GoTO may be helpful for readers. 

This manuscript is a resubmission of an earlier submission. The following is a list of the peer review reports and author responses from that submission.

Round 1

Reviewer 1 Report

This research is interesting and relevant, it will be much improved when the following suggestions are made.

Introduction:

The definition of GoTO should be made in the introduction section, and the following sentence should be moved forward to introduction rather than discussion:

“The process navigator has been named GoTO (Goal to Outcome) because it helps the involved actors to navigate through the identification of goals, it ensures awareness and ownership among the involved actors, and it supports the alignment of steps and processes with a trajectory towards the expected outcomes.”

Reference 1-8, some of them are too old to be updated and the sources of referencing should be from more recognized journals, such of Journal of medical internet research health affairs etc...regarding the reference of health intervention can refer to the following articles:

Kunonga T, Spiers G, Beyer F, Hanratty B, Boulton E, Hall A, Bower P, Todd C, Craig D

Effects of Digital Technologies on Older People’s Access to Health and Social Care: Umbrella Review

J Med Internet Res 2021;23(11):e25887  https://www.jmir.org/2021/11/e25887

Structure:

Try to avoid small paragraphs and they should be integrated to a more representative paragraph. Eg.

“Small and middle-sized health and care organisations (SMHO) with less resources 54

which may result in lack of awareness of previous results and unwillingness to face and tackle organisational challenges, may find it difficult to initiate and implement new solutions [14]”.

This paragraph should be developed with possible solutions, such as stakeholder management of other management methods.

This can refer to the following articles:

Chen S, Liu C, Wang Z, McAdam R, Brennan M, Davey S, Cheng T

How Geographical Isolation and Aging in Place Can Be Accommodated Through Connected Health Stakeholder Management: Qualitative Study With Focus Groups

J Med Internet Res 2020;22(5):e15976 https://www.jmir.org/2020/5/e15976.

*Terminology,

The terminology should stay consistence and refer to the existing literature.

Eg, screen consultation can be replaced by tele-conference and the term of physical distancing world-wise should be reconsidered as well…etc…

For the terminology consistence can refer to the following articles”

Chen, S.C.-I.; Liu, C.; Hu, R.; Mo, Y.; Ye, X. “Nomen Omen”: Exploring Connected Healthcare through the Perspective of Name Omen. Healthcare 2020, 8, 66. https://doi.org/10.3390/healthcare8010066.

Discussion:

Research strength and limitations as well as future research agenda should be added in the end of this section.

References:

The quality and of the sources should be taken into account. Also try to keep the references up-to-dates.

Authors reply:

This research is interesting and relevant, it will be much improved when the following suggestions are made.

Introduction:

The definition of GoTO should be made in the introduction section, and the following sentence should be moved forward to introduction rather than discussion:

“The process navigator has been named GoTO (Goal to Outcome) because it helps the involved actors to navigate through the identification of goals, it ensures awareness and ownership among the involved actors, and it supports the alignment of steps and processes with a trajectory towards the expected outcomes.”

Answer: Thank you for this suggestion, we have added the sentence to the introduction.

Reference 1-8, some of them are too old to be updated and the sources of referencing should be from more recognized journals, such of Journal of medical internet research health affairs etc...regarding the reference of health intervention can refer to the following articles:

Kunonga T, Spiers G, Beyer F, Hanratty B, Boulton E, Hall A, Bower P, Todd C, Craig D Effects of Digital Technologies on Older People’s Access to Health and Social Care: Umbrella Review . J Med Internet Res 2021;23(11):e25887  https://www.jmir.org/2021/11/e25887

Answer: Thank you for this article. We have now updated the references, but we find it important to keep some of the strategic reports such as meta reviews and policy papers, as they represent an important perspective to cover in this area since not all is documented in research articles. We have removed Pare, 2007.

Structure:

Try to avoid small paragraphs and they should be integrated to a more representative paragraph. Eg.

“Small and middle-sized health and care organisations (SMHO) with less resources 54 which may result in lack of awareness of previous results and unwillingness to face and tackle organisational challenges, may find it difficult to initiate and implement new solutions [14]”.

This paragraph should be developed with possible solutions, such as stakeholder management of other management methods.

This can refer to the following articles: Chen S, Liu C, Wang Z, McAdam R, Brennan M, Davey S, Cheng T. How Geographical Isolation and Aging in Place Can Be Accommodated Through Connected Health Stakeholder Management: Qualitative Study With Focus Groups. J Med Internet Res 2020;22(5):e15976 https://www.jmir.org/2020/5/e15976.

Answer: Thank you for pointing this out – we have rephrased the paragraph so the link to the previous paragraph on larger organisations is clearer.

Terminology:

The terminology should stay consistence and refer to the existing literature.

Eg, screen consultation can be replaced by tele-conference and the term of physical distancing world-wise should be reconsidered as well…etc…

For the terminology consistence can refer to the following articles” Chen, S.C.-I.; Liu, C.; Hu, R.; Mo, Y.; Ye, X. “Nomen Omen”: Exploring Connected Healthcare through the Perspective of Name Omen. Healthcare 2020, 8, 66. https://doi.org/10.3390/healthcare8010066.

Answer: We understand and appreciate this comment. However, currently there is an on-going digital health service transformation and the terms used exist in many countries (at least in Northern Europe) as a natural part of the language and most are actually better than many of the established terms. We disagree that screen consultation can be replaced with tele-conference. With regards to physical distancing we appreciate that a lot of the Anglo literature use “social distance” however in most non-anglo contexts “social” refers to a cognitive aspect and therefore is not considered suitable in the discussion of THC solutions.

Discussion:

Research strength and limitations as well as future research agenda should be added in the end of this section.

Answer: This has now been added.

References:

The quality and of the sources should be taken into account. Also try to keep the references up-to-dates.

Answer: thank you for this comment, we have added some references and deleted some in accordance with your previous remarks. 

Reviewer 2 Report

Overall impression: The topic of this article is very interesting and significant to healthcare practice. Especially with COVID-19 more healthcare services are provided remotely and supporting small and middle-sized healthcare organizations in implementing telehealth options is very important. Yet, the study has major flaws dampening my enthusiasm for this article. The methods used are insufficiently described preventing a thorough evaluation of the presented results. Further, the first part of the results present a summary of existing HTA and project models without sufficient synthesis of model components to guide an evidence-based development of the new model.

Reviewer comments by section:

Introduction

  • The importance for telehealth options for both patients and organizations is very well highlighted in the first part of the introduction section. 
  • The introduction section does not provide a comprehensive overview of challenges and issues for telehealth implementation. In addition to limited resources (lines 54-57), please elaborate on other challenges such as reimbursement issues, regulatory issues (e.g., HIPAA laws, or EU data protection laws), limited technological infrastructure within organizations, provider-side issues (e.g., low digital literacy, issues adjusting to new healthcare delivery mode), etc. 
  • Changing objective/assumptions in the latter part of the introduction section are confusing. The authors would benefit from a cohesive presentation of the final goals and objectives to conduct the study. Please revise this section.

Methods

  • Article refers to Master's thesis of one of the authors which is not included in the references. All used work has to be cited. If the Master's thesis is published locally (e.g., university library) it must be included as grey literature resource. If the Master's thesis is not published, it still must be included as unpublished work. Please include the citation in the appropriate format.
  • Conversations with experts as part of the narrative reviews are not sufficiently described. Please provide information on a) how many experts were included; b) how experts were identified/chosen; c) what questions were asked (best practice: include the conversation guide as supplemental material).
  • Figure 1 depicts well how HTA-related models were identified.  Please also describe how the project and implementation models were identified. 
  • Stage 3: The reasoning for the selection criteria is insufficient described. Please provide more details explaining the importance of each of the selection criteria.

Results

  • Second part of Research question 1 ("... how can these core components be combined to inform a process navigator?") and Research question 2 are not answered in the results section as the results section solely presents the GoTO navigator. Please provide detailed information on the reasoning of selecting the parts and steps of GoTO.
  • Subsection 3.1. to 3.13. summarize existing models. Synthesis of model components is limited to a short paragraph (4. Summary). Sections 3.1. to 3.13. should be a table or supplemental material as it is all described in the cited literature. Please focus and expand the synthesis of the model components and how it informs GoTO.
  • Subsection 4.1. -- Please provide more detail of how these steps were chosen and compiled, and how it links back to the defined section criteria in the methods section. As of now the parts and steps of the GoTO tool seem arbitrarily chosen.

Discussion

  • The discussion section raises some important topics and limitations of this study.
  • Lines 649-652: Please elaborate more on the improvements of GoTO compared to CSE. It is crucial to discuss in detail how the existing CSE model is expanded.
  • Limiting this study to HTA-related, project and innovation models is a major flaw. There are many implementation models and frameworks in the healthcare literature (overview can be found here: https://impsciuw.org/implementation-science/research/frameworks/) that could have informed GoTO. To improve current practices and present a superior tool for telehealth development and implementation, a comprehensive review of relevant tools and frameworks is crucial and highly recommended. 

Conclusion

  • The conclusion is based on the authors judgement of the GoTO tool and does not link back to components of existing models or other robust evaluation criteria.

Figures and Abbreviations

  • Figures must include a legend explaining all abbreviations used
  • Please include all abbreviations in the text and list of abbreviations. For example, BioX is not explained.

Authors reply:

Overall impression: The topic of this article is very interesting and significant to healthcare practice. Especially with COVID-19 more healthcare services are provided remotely and supporting small and middle-sized healthcare organizations in implementing telehealth options is very important. Yet, the study has major flaws dampening my enthusiasm for this article. The methods used are insufficiently described preventing a thorough evaluation of the presented results. Further, the first part of the results present a summary of existing HTA and project models without sufficient synthesis of model components to guide an evidence-based development of the new model

Answer: We acknowledge the reviewer’s position and fully understand that our work cannot be termed evidence based. We will not claim that it is a research driven or even informed process. It should rather be considered an opinion paper. The initial work was completed in relation to a field study where a large telehealth project tilted and the author who followed this project realized that they lacked a good and easy model to assist them to avoid this in the future. Thus, she started the work on GoTO. We, a group of health informaticians, then reviewed her thesis work and expanded on it and had several discussions with colleagues in other countries. Based on the identified literature and conversations we proposed the GoTO model. Today it is used by our students and in an EU project. It is meaningful and logic and easy to use without certification etc. We are therefore convinced that the publication and dissemination of the model will help many to understand the key principles in an innovation and implementation process. We also underline that our proposal is not of high originality as the model builds on stitching together known models. In any case, we know from conversations that this publication will be of value to many.

We think that the above-mentioned details may help to understand the reason for this publication and the main points are already in the article, so we have not changed anything in relation to this comment, but hope the reviewer will acknowledge our intentions with this paper.

Reviewer comments by section:

Introduction:

The importance for telehealth options for both patients and organizations is very well highlighted in the first part of the introduction section. 

The introduction section does not provide a comprehensive overview of challenges and issues for telehealth implementation. In addition to limited resources (lines 54-57), please elaborate on other challenges such as reimbursement issues, regulatory issues (e.g., HIPAA laws, or EU data protection laws), limited technological infrastructure within organizations, provider-side issues (e.g., low digital literacy, issues adjusting to new healthcare delivery mode), etc. 

Answer:  These many important aspects have been addressed in our proposal of GoTO and in particular in the later work with developing a guide for the first two parts of GoTO which is referenced in the discussion. We have now added two references to elaborate a little on some of the challenges. We have added two references; a review condensing the success and failure factors in eHealth and a reference to the Normalisation process theory important steps for success. 

Changing objective/assumptions in the latter part of the introduction section are confusing. The authors would benefit from a cohesive presentation of the final goals and objectives to conduct the study. Please revise this section.

Answer: Thank you for pointing this out; this has now been done and it will help to read our assumptions that has underpinned the selection process of elements from the various models.

Methods

Article refers to Master's thesis of one of the authors which is not included in the references. All used work has to be cited. If the Master's thesis is published locally (e.g., university library) it must be included as grey literature resource. If the Master's thesis is not published, it still must be included as unpublished work. Please include the citation in the appropriate format.

Answer: The thesis is now referred to as unpublished work and in a later paragraph in the methods section, we have now cited an abstract from a conference in a PubMed indexed journal as a reference to the initial work. (Reference 35)

Conversations with experts as part of the narrative reviews are not sufficiently described. Please provide information on a) how many experts were included; b) how experts were identified/chosen; c) what questions were asked (best practice: include the conversation guide as supplemental material).

Answer: this was informal conversations and structured by templates. It has taken place over the years and taken place in projects. We have now added countries of the colleagues, but are not able to add more details.

Figure 1 depicts well how HTA-related models were identified.  Please also describe how the project and implementation models were identified. 

Answer: They were, as already described, identified in a process where we drew upon knowledge and experience by the authors and discussions with colleagues in our network.

Stage 3: The reasoning for the selection criteria is insufficient described. Please provide more details explaining the importance of each of the selection criteria.

Answer: We have now expanded on this but also find that it is already described in section 4 of the article.

Results

Second part of Research question 1 ("... how can these core components be combined to inform a process navigator?") and Research question 2 are not answered in the results section as the results section solely presents the GoTO navigator. Please provide detailed information on the reasoning of selecting the parts and steps of GoTO.

Answer:
We have now removed the word hypothesis and the research questions from the introduction to make the process clearer. The synthesis of the GOTO is described with references to the elements in section 4. We have also changed the presentation of the models to tables to ease the reading. We hope this clarifies how we together with the criteria in the methods section reached the result of the GOTO model.

Subsection 3.1. to 3.13. summarize existing models. Synthesis of model components is limited to a short paragraph (4. Summary). Sections 3.1. to 3.13. should be a table or supplemental material as it is all described in the cited literature. Please focus and expand the synthesis of the model components and how it informs GoTO.

Answer: We have now changed to tables as suggested.

Subsection 4.1. -- Please provide more detail of how these steps were chosen and compiled, and how it links back to the defined section criteria in the methods section. As of now the parts and steps of the

Answer: We have now tried to edit this, but find that it already has been described. To ease the reading we have now changed the format of results into tables so the description of the synthesis of the models read better.

Discussion

The discussion section raises some important topics and limitations of this study.

Lines 649-652: Please elaborate more on the improvements of GoTO compared to CSE. It is crucial to discuss in detail how the existing CSE model is expanded.

Limiting this study to HTA-related, project and innovation models is a major flaw. There are many implementation models and frameworks in the healthcare literature (overview can be found here: https://impsciuw.org/implementation-science/research/frameworks/) that could have informed GoTO. To improve current practices and present a superior tool for telehealth development and implementation, a comprehensive review of relevant tools and frameworks is crucial and highly recommended. 

Answer: Thanks for this input. We have now added this to a section with limitations. We agree that a newer review of the field would be of benefit. The frameworks we have included are primarily used for development of telehealth solutions or have been used in this area to a large extend. Many of the other frameworks are more general. We fully acknowledge that it is a flaw that we have not been aware of some relevant frameworks when we started our work, however, we find that the overall principles are still the same. We could have benefitted from having addressed earlier definitions, but on the other hand, the whole idea is to communicate a model which can be used by non-academic people in real-life settings working by themselves or in small groups prior to starting telehealth based projects. 

Conclusion

The conclusion is based on the authors’ judgement of the GoTO tool and does not link back to components of existing models or other robust evaluation criteria.

Answer: Yes, we consider a conclusion to be a short text about the product.

Figures and Abbreviations

Figures must include a legend explaining all abbreviations used

Answer: Thank you for this comment; considering the amount of abbreviations used and the fact that these are familiar to most readers, we have inserted a reference to section 8. List of abbreviations.

Please include all abbreviations in the text and list of abbreviations. For example, BioX is not explained.

Answer: The abbreviation list has been checked and revised. BioX is not an abbreviation but a name, we did however find some inconsistencies in how the Standford BioX design innovation process is referenced and this has now been corrected.